

# A taxonomically and geographically constrained information base limits non-native reptile and amphibian risk assessment: a systematic review

Nicola J. van Wilgen[1,2], Micaela S. Gillespie[2], David M. Richardson[2] and John Measey[2]

[1] Cape Research Centre, South African National Parks, Steenberg, Western Cape, South Africa
[2] Centre for Invasion Biology, Department of Botany & Zoology, Stellenbosch University, Matieland, South Africa

## ABSTRACT

For many taxa, new records of non-native introductions globally occur at a near exponential rate. We undertook a systematic review of peer-reviewed publications on non-native herpetofauna, to assess the information base available for assessing risks of future invasions, resulting in 836 relevant papers. The taxonomic and geographic scope of the literature was also compared to a published database of all known invasions globally. We found 1,116 species of herpetofauna, 95% of which were present in fewer than 12 studies. Nearly all literature on the invasion ecology of herpetofauna has appeared since 2000, with a strong focus on frogs (58%), particularly cane toads (*Rhinella marina*) and their impacts in Australia. While fewer papers have been published on turtles and snakes, proportionately more species from both these groups have been studied than for frogs. Within each herpetofaunal group, there are a handful of well-studied species: *R. marina*, *Lithobates catesbeianus*, *Xenopus laevis*, *Trachemys scripta*, *Boiga irregularis* and *Anolis sagrei*. Most research (416 papers; 50%) has addressed impacts, with far fewer studies on aspects like trade (2%). Besides Australia (213 studies), most countries have little location-specific peer-reviewed literature on non-native herpetofauna (on average 1.1 papers per established species). Other exceptions were Guam, the UK, China, California and France, but even their publication coverage across established species was not even. New methods for assessing and prioritizing invasive species such as the Environmental Impact Classification for Alien Taxa provide useful frameworks for risk assessment, but require robust species-level studies. Global initiatives, similar to the Global Amphibian Assessment, using the species and taxonomic groups identified here, are needed to derive the level of information across broad geographic ranges required to apply these frameworks. Expansive studies on model species can be used to indicate productive research foci for understudied taxa.

Corresponding author
Nicola J. van Wilgen,
nvanwilgen@gmail.com

## INTRODUCTION

Alien or non-native species are taxa that have been transported beyond the limits of their natural range, and may become invasive in new areas if they are able to form established, self-sustaining populations in these new locations (*Blackburn et al., 2011*). For several taxa, new invasions continue to occur at a near exponential rate (*Seebens et al., 2017*). Evidence of the negative impacts of many invasive species (*Pimentel, 2011*), including impacts from reptiles and amphibians (*Shine, 2014*; *Kraus, 2015*; *Measey et al., 2016*) is increasing. This has added urgency to the pursuit of achieving a thorough understanding of factors mediating success at different stages of the introduction-naturalisation-continuum (*Richardson et al., 2000*; *Blackburn et al., 2011*) to inform policies to reduce the risk of further invasions. Effective and defensible policies are increasingly being introduced, or considered, to attempt to curb invasions at a national level (*Genovesi et al., 2015*), but require robust models to differentiate between innocuous and potentially problematic species (*Springborn, Romagosa & Keller, 2011*). The challenges posed to the assessment of invasion risk are unique to each taxonomic group (*Kumschick & Richardson, 2013*). Such challenges include strategically prioritising research on the impacts of understudied species, in areas of the world where invasive species are less studied and different aspects of the invasion process less well covered. Although fairly robust models exist to explain the success of introduced plants (*Pheloung, Williams & Halloy, 1999*), formal risk assessment still faces multiple challenges even for this group (*Hulme, 2012*; *Speek et al., 2013*).

Invasions of reptiles and amphibians ('herpetofauna'), although not nearly as well documented as plant invasions (*Pyšek et al., 2008*), have received significant attention in recent years, especially in Florida (*Krysko et al., 2011*), with an emphasis on species in the pet trade and 'hitchhiker' species (*Tingley et al., 2018*). While there are currently relatively few invasive reptiles and amphibians, some species do have significant impacts (*Kraus, 2015*; *Measey et al., 2016*) and records of first introduction for reptiles at least, have been increasing at an exponential rate since the 1950s (*Seebens et al., 2017*). Numerous studies have investigated the factors that influence the popularity in the pet trade, probability of introduction and release (*Stringham & Lockwood, 2018*), and the likelihood of successful establishment of herpetofaunal species in new regions. Several strong and consistent patterns have emerged from these studies. For example, establishment success is enhanced for species introduced and released in high numbers (*Garcia-Diaz et al., 2015*; *Mahoney et al., 2015*), into areas with high native species richness (*Tingley, Phillips & Shine, 2011*; *Ferreira et al., 2012*; *Poessel et al., 2013*), for species that are less manageable (more expensive to keep, prone to escape, aggressive or venomous, *Fujisaki et al., 2009*), that have fast-paced life histories (*Allen et al., 2017*; *Van Wilgen & Richardson, 2012*), and that come from areas with similar environmental conditions to the area of release (*Bomford et al., 2009*; *Van Wilgen, Roura-Pascual & Richardson, 2009*; *Tingley, Phillips & Shine, 2011*; *Rago, While & Uller, 2012*). However, models using these predictors, can be relatively data hungry, making it difficult to regulate the import of species and prioritise management of those already present in an

area in the absence of primary data. This is increasingly leading to disillusionment among traders who challenge restrictions placed on the importation of certain taxa and could spur illegal imports. Increasing objectivity and a better understanding of uncertainties are necessary to inform communication between legislators, managers, conservationists and pet traders.

Formal pre-border risk assessment, which is increasingly becoming mandatory in many countries to regulate the import of species that are not yet present, faces important challenges. Risk is defined by both the probability of an event taking place and the consequences of such an event (*Kumschick & Richardson, 2013*). Most risk assessments for non-native reptiles and amphibians to date have focused on the potential for a species to establish (event probability). Another approach recently proposed is to assess the impacts of a non-native species based on known impacts in other parts of its non-indigenous range (EICAT: *Blackburn et al., 2014*; *Hawkins et al., 2015*; SEICAT: *Bacher et al., 2018*). This approach, and others like it (*Kumschick et al., 2015*; *Nentwig et al., 2016*), attempt to provide an index to denote the likelihood of impact if a species is introduced into a new location, for example through trade. Managers could use these simple indices to inform decision-making. Unlike most risk-assessment protocols, EICAT and SEICAT rely exclusively on a systematic review of peer-reviewed publications of each species in their non-indigenous range. This 'gold standard' of data sources carries with it the need for a comprehensive set of literature on all non-native species, but assessments to date have highlighted problems with the assumption that sufficient published material exists. For example, an assessment of all non-native bird populations found published material detailing ecological impact for only 30% of 415 species (*Evans, Kumschick & Blackburn, 2016*). A similar assessment for non-native amphibians found information on ecological impact for only 38% of 105 species (*Kumschick et al., 2017*), and socio-economic information for only 7% of these species (*Measey et al., 2016*; *Bacher et al., 2018*). It would be naive to assume that such biases result from an absence of impacts; it is well known that reporting biases exist between different taxa, and between continents/global regions (*Dawson et al., 2017*). However, until the extent of biases are better understood, it remains difficult to motivate studies on non-native species in poor or underdeveloped localities, even though such data may be of great value to more developed nations who have legal and logistical infrastructure to prevent importation.

We aim to assess the information currently available to inform quantitative risk assessments of herpetofauna and the degree of taxonomic and geographic bias that exists in the literature. To do this, we performed a systematic review of research published to date on non-native reptile and amphibian species to assess the scope of peer-reviewed information currently available. In so doing, we highlight the species, subjects and geographic locations that received the most research, and enable identification of gaps.

## SURVEY METHODOLOGY

For this analysis we split reptiles and amphibians into six taxonomically-based 'morphological groups' (hereafter herpetofaunal groups): lizards, snakes, turtles, crocodiles, frogs and salamanders, recognizing that snakes and lizards are not separate

monophyletic groups. However, this distinction is reflective of the level at which these groups are often studied (*Rodda et al., 1999*) as a result of their distinct morphological characteristics and different selection pressures on these groups in the pet trade that result in divergent invasion patterns as well as other functional differences that are likely to affect their invasion probability.

To determine the knowledge base underpinning existing risk assessments for herpetofauna, we reviewed the literature available on the introduction, establishment and/or invasion and impact of non-native reptiles and amphibians. Searches were conducted on the ISI Web of Science Core Collection (on 3 March 2016) using the following criteria: Topic = alien OR invasive OR non-native OR exotic OR non-indigenous OR feral AND Topic = reptil* OR amphibia* OR turtle* OR tortoise* OR lizard* OR herpetofauna OR crocod* OR anura OR caudata OR testudin* OR ophidia OR sauria OR squamata OR snake* OR frog* OR toad* OR salamand* OR newt*. The ISI subscription used literature dating back to 1970. We recognise that ISI is biased in many respects, including against non-English literature (*Adam, 2002*). However, we justify the use of this database alone as it is a source likely to be used by those conducting risk assessments, and for whom non-English language content may also be inaccessible. This search yielded 3,194 papers. Many of these papers were not relevant to the current study as a result of a number of homonyms (e.g. invasive and non-invasive medical techniques) that resulted from including the wide range of search terms (*Westgate & Lindenmayer, 2017*). As a preliminary measure to reduce the papers from extraneous study fields (*Westgate & Lindenmayer, 2017*), the results were refined by excluding irrelevant research areas (e.g. paediatrics and ophthalmology, see the full list in supplementary material), leaving us with 2,383 papers (Fig. 1). The relevance of these papers was assessed by reading the abstract and, where necessary, the full paper. All studies that pertained to non-native herpetofaunal species were included in the review. These included those that addressed any aspect of the invasion process (e.g. establishment or potential establishment correlates, or bioclimatic models); impacts or potential impacts (including ecological impacts, human socio-economic impacts, spread of disease and envenomation); potential or actual spread dynamics; genetics of introduced populations; detectability; ecology in non-native range (diet, reproduction, interactions with other species); biology in non-native range; and factors relating to import or pathways of invasion, including trade in non-natives and translocation of non-natives; or first records of populations. Studies focussing on the impacts of non-herpetofaunal non-native species (e.g. mammals or fish) on native reptiles and amphibians were excluded, as were purely veterinary studies, studies on parasites of non-native species where the parasite and not the non-native species was the focus, and any papers that had no clear link to the topics of interest (e.g. despite the filter on the research areas, the search identified a large number of papers from other disciplines, such as invasive vs non-invasive medical techniques). This final list included 836 papers that were used in the quantitative analysis.

For each paper, the identity of all non-native reptiles and/or amphibians included in the work was recorded, as was the country or US state (locality) in/across which the

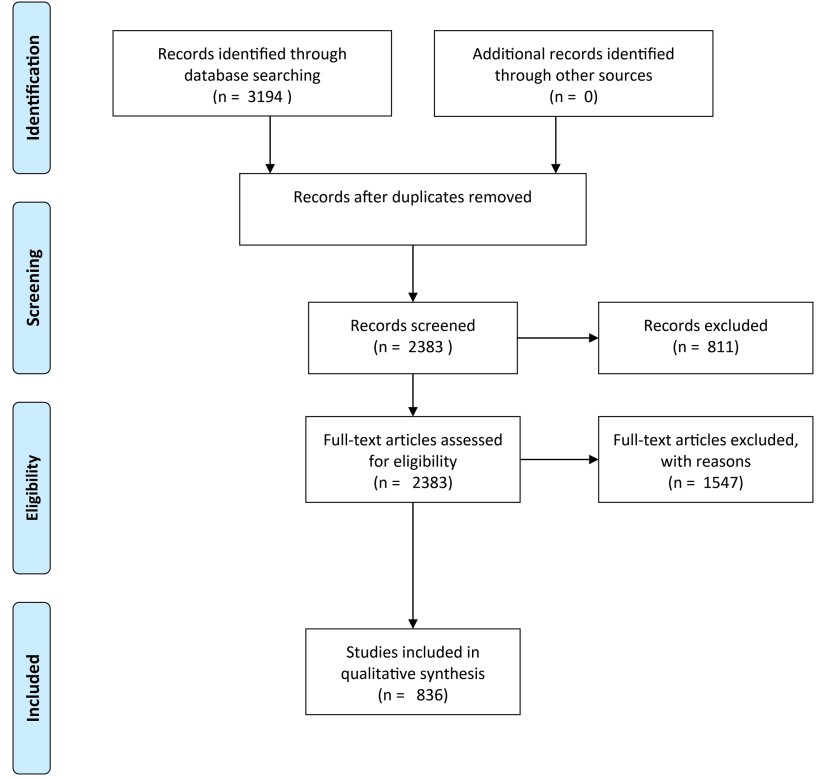

**Figure 1 Prisma flowchart.** Prisma flow diagram (*Moher et al., 2009*) for systematic review of articles on invasive amphibians and reptiles from the Web of Science (formerly Science Citation Index) on 3 March 2016. Search criteria used: Topic = alien OR invasive OR non-native OR exotic OR non-indigenous OR feral AND Topic= reptil* OR amphibia* OR turtle* OR tortoise* OR lizard* OR herpetofauna OR crocod* OR anura OR caudata OR testudin* OR ophidia OR sauria OR squamata OR snake* OR frog* OR toad* OR salamand* OR newt*.   

study took place. Species recorded from papers included those that were the direct focus of study (e.g. for which field data were collected or species that were included in an experimental setup) and any species documented in a particular place outside of their native range. We recognise that detailed studies of single species are probably more useful to risk assessors. However, papers that provide general overviews of many species are also valuable (with specified confidence limitations) for risk assessment when specific information about focal species is not available, and also to provide evidence of pathways and introduction records, where introduction represents an important phase of invasion (*Blackburn et al., 2011*). Each paper was also classed into one or more of the following subject categories: Climate, Impacts, Pathways, Control, Invasion Correlates, Distribution, Translocation, or Trade. Once all data were captured, the taxonomy of species was cross-checked using *Frost (2017)* for amphibians and *Uetz, Freed & Hošek (2017)* for reptiles, and relevant entries in the database were consolidated. The family to which each species belongs, and the total size of these families, and herpetofaunal groups, were taken from the same sources. Many papers that covered more than one taxon included species that were non-native, but had not necessarily been released into the wild, or did not have established populations. We did not attempt to remove these species from

our list, which leads to discrepancies in numbers between our dataset and those of *Kraus (2009)*. We did however compare results for papers that dealt only with one species with those that dealt with multiple species as well as the full set of papers, but found no difference in the subject of these subsets of publications. The full dataset is available as Supplementary Material S2. Summary statistics were calculated by paper, subject and species.

We assessed taxonomic biases in the literature at family level by comparing the number of species per reptile or amphibian family present in the reviewed literature with a random expectation generated using the hypergeometric distribution (*Van Wilgen et al., 2010*) in R v. 3.4.0 (*R Core Development Team, 2017*). The hypergeometric distribution is similar to a binomial distribution and describes the probability of a given number of successes given a specified number of draws, without replacement. In this instance, a set number of species are sampled from a pool of families of known size. Families outside the 95% confidence intervals were deemed to be either over- or under-represented in the literature, compared to expectations based on the size of the family and the total number of species that appear in the literature. Results were visualised by plotting the number of species that have been described globally within each family against the proportion of species in the literature under review. We also performed the same analysis at the level of herpetofaunal groups.

Taxonomic bias was further assessed at herpetofaunal-group level by comparing information on known invasions published in *Kraus (2009)*, to information available from peer-reviewed literature (this review). To provide a baseline of taxonomic and geographic scope of known reptile and amphibian invasions, we used data from *Kraus (2009)*, a database that details all known introductions of non-native herpetofauna from the published and grey literature at the time (2006/2007). While these data are somewhat out of date, they represented the latest comprehensive dataset of introduced and established species across taxa at the time and for the purposes of our analysis we assume that derived ratios and trends will have remained similar. Populations in the *Kraus (2009)* database are recorded at country, island or US-state level (hereafter location). Any (or multiple) successful population(s) within a country or state is counted as a successful introduction for that location (and the total number of introductions or populations within a particular country or US state is not considered). We refer to 'introduction' as the arrival, outside of captivity, of a species in an area where it is not native (i.e. having overcome a natural barrier to movement, *Blackburn et al., 2011*). Successful introductions (established species) were classified according to *Kraus (2009)* as introductions 'reported to be established (within the country, island or US state) at the time of the most recent literature citation for the population in question'. 'Established' is interpreted to mean a population that has shown the ability to reproduce regularly, without human intervention (or in spite of human interventions) to form populations of sufficient size to be resilient to stochastic events. From *Kraus (2009)*, we extracted the total number of species introduced anywhere outside of their native range and the total number of known independent introductions per species per location (some species may have been introduced to more than one country or US state) for each herpetofaunal

group. We then calculated the proportion of successfully established populations and species as a fraction of those introduced and as a fraction of all described species within the group.

Potential geographic bias in the literature on non-native herpetofauna identified in this review was assessed at a crude level by comparing the number of studies for each country or US state with the number of successfully established species in that country (*Kraus, 2009*). For multi-location studies, we scored only papers that truly collected/ provided data for all localities defined within a region, and excluded studies which did not have a defined geographic scope or had clear geographical bias through selection of species (*Kats & Ferrer, 2003*; *Pilliod, Griffiths & Kuzmin, 2012*, where geographic scope was restricted by the example species selected) or areas (e.g. regional studies on reptile trade, such as *Rataj et al., 2011* or *Herrel & Van Der Meijden, 2014*, where information was clearly not sourced from all countries or locales). A total of 800 papers for which exact country locations could be identified were included in this analysis. This included 22 papers deemed to be global in scope, for which all countries and US states were scored.

We also conducted a more detailed assessment of geographic coverage, extracting the species lists and locations from each paper and tallying the total number of species covered by each paper per location (country or state). The number of species studied was compared to the number of species established per location (*Kraus, 2009*). Multi-location or global papers that dealt with more than one species were only included in this analysis in instances where it was possible to assign all the species in the paper to particular locations. Papers such as *Bomford et al. (2009)* were therefore excluded as the paper did not specify whether the species used in the models had been introduced to Britain, Florida or California. A total of 767 papers were included in this analysis. All analyses were conducted in R v. 3.4.0 (*R Core Development Team, 2017*), and maps produced in ARCMap 10.3.1 (*ESRI, 2015*).

## RESULTS

### Representation of herpetofaunal groups in the literature

Only 1,116 species (6% of c. 18,145 described herpetofaunal species; *Frost, 2017*; *Uetz, Freed & Hošek, 2017*) are included in the 836 studies relating to herpetofaunal invasions in this review. A total of 552 (49%) of these species appear in only one study, and 909 species (81%) appear in fewer than five studies. A total of 95% of species were included in fewer than 12 studies. Hardly any studies were conducted on non-native herpetofauna before 1990, and most studies were published after 2000 (Fig. S1). A large proportion of papers (653/836, 78%) focused on a single species, and only a minority (54 or 6%) reported on more than 10 species.

Of the papers included in this review, most focus on or include frogs (58%, Table 1), while lizards appear in almost a quarter of studies. The remaining herpetofaunal groups are represented in 18% or fewer of papers relating to non-native reptiles and amphibians. However, frogs and lizards are the largest herpetofaunal groups and as such, the number of species from these groups present in the literature actually under-represents

**Table 1 Representation of groups of reptiles and amphibians across papers from the systematic review.**

| Group | Number of papers (out of 836) |
| --- | --- |
| Crocodiles | 22 (3%) |
| Frogs | 487 (58%) |
| Lizards | 198 (24%) |
| Salamanders | 33 (4%) |
| Snakes | 149 (18%) |
| Turtles | 131 (16%) |

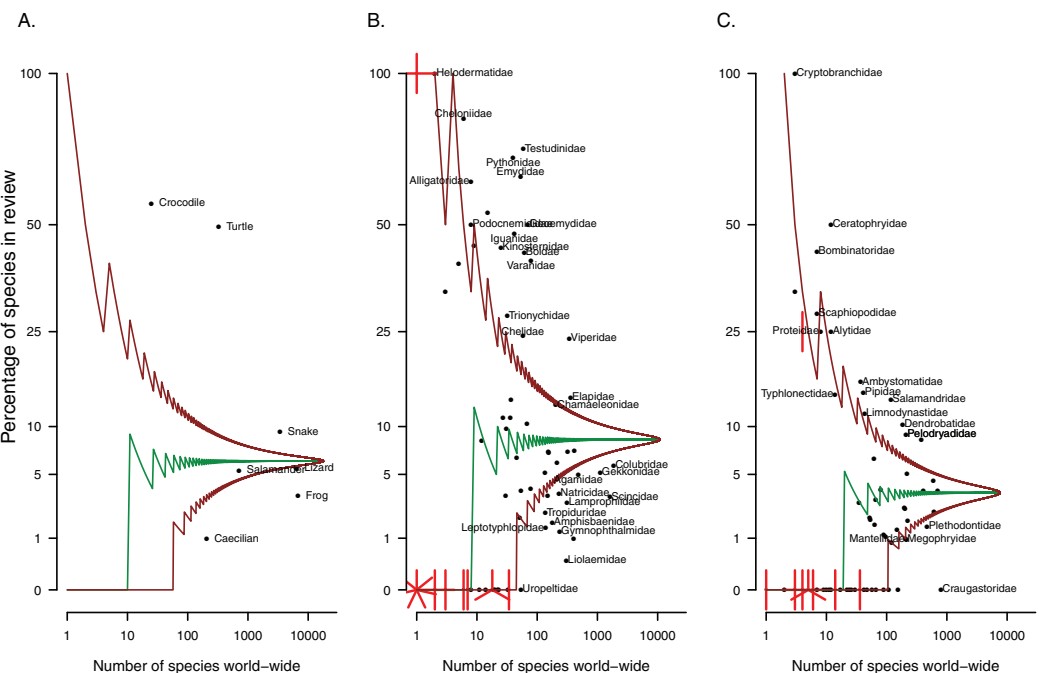

**Figure 2 Patterns in taxonomic representation of (A) herpetofaunal groups and (B) reptile and (C) amphibian families present in the invasion ecology literature.** The median (middle green line) and 95% confidence intervals (brown lines), adjusted for multiple comparisons, were estimated from the hypergeometric distribution. The points represent herpetofaunal groups or families; those that fall between the brown lines are not significantly over or under-represented (relative to amphibians or reptiles as a whole). Where multiple points overlap, lines indicate the number of points at each location.

these groups as a whole (Fig. 2A). Turtles and crocodiles are comparatively small herpetofaunal groups (25 crocodile species and 347 testudine species have been described to date) and have the most representative sample of species covered by the literature. Both groups are overrepresented in the papers that were reviewed (Fig. 2A, 47% and 56% of species from these respective groups occur in at least one of the papers included in this review, Table 2), as are snakes.

Some families have received more attention than others. For reptiles, nearly all the testudine and crocodilian families (e.g. Cheloniidae, Testudinidae, Emydidae, Alligatoridae, Crocodylidae) are overrepresented in the literature, that is they have

**Table 2 Success rates for introductions and species in each herpetofaunal group (lizards, crocodiles, snakes, turtles, frogs and salamanders) as per Kraus (2009).**

| Order | Total species Frost (2017) and Uetz, Freed & Hošek (2017) | Number of species included in our review (% of total species described per herpetofaunal group) | Introduction success | | | Species success | | | Proportion of taxonomic sampling | |
|---|---|---|---|---|---|---|---|---|---|---|
| | | | Total introductions Kraus (2009) | Successful introductions Kraus (2009) | Success rate of introductions Kraus (2009) (%) | Total species introduced Kraus (2009) (approximate) | Successful species Kraus (2009) | Success rate of species Kraus (2009) (%) | Proportion of species introduced outside of native range (%) | Proportion of species naturalised/invasive outside of native range (%) |
| Lizards | 6,459 | 347 (5.4%) | 716 | 445 | 62 | 243 | 139 | 57 | 3.8 | 2.2 |
| Crocodiles | 25 | 14 (56%) | 29 | 3 | 10 | 7 | 1 | 14 | 28 | 4.0 |
| Snakes | 3,619 | 320 (8.8%) | 370 | 115 | 31 | 139 | 30 | 22 | 3.8 | 0.8 |
| Turtles | 347 | 162 (46.7%) | 423 | 147 | 35 | 93 | 40 | 43 | 26.8 | 11.5 |
| Frogs | 6,776 | 223 (3.3%) | 508 | 313 | 62 | 147 | 82 | 56 | 2.2 | 1.2 |
| Salamanders | 713 | 38 (5.3%) | 81 | 35 | 43 | 37 | 19 | 51 | 5.2 | 2.7 |
| Caecilians | 206 | 2 (1%) | | | | | | | | |

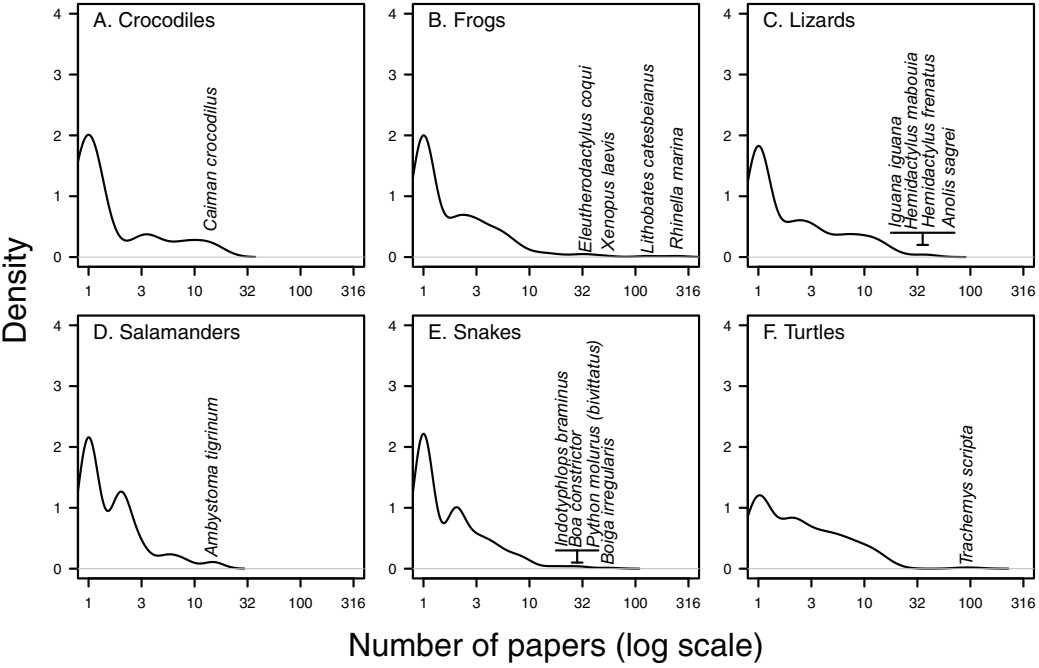

Figure 3 **Kernel density of papers per species.** Density plots of the number of Web of Science papers per species in each herpetofaunal group (A. Crocodiles, B. Frogs, C. Lizards, D. Salamanders, E. Snakes and F. Turtles) show that the majority of species feature in only one paper, while there are generally a few species that appear in a high number of papers. Taxa featuring in the highest number of papers have been highlighted for each group.                               

more species written about than expected by chance given the size of the family, compared to reptiles as a whole (Fig. 2B). Snake families that stand out include boids (Boidae and Pythonidae), Viperidae and Elapidae, while the lizard families most overrepresented across studies are iguanids (Iguanidae and Corytophanidae), varanids and chameleons. Over-represented amphibian families include three of the nine salamander families (Ambystomatidae, Cryptobranchidae and Salamandridae), and nine out of 56 frog families, including Ranidae, Pipidae, Dendrobatidae, Ceratophryidae and Bombinatoridae (see Fig. 2C for these and others). Families might be under-represented in the literature if (1) species in these families are being translocated without being reported, or if (2) species in these families are genuinely not moved around, typically resulting in no or very few non-native representatives for the group. The fleshbelly frogs (Craugastoridae, approximately 800 species) were the largest amphibian family with no species present in the invasion ecology literature. The largest reptile family with no representatives was the Uropeltidae (shieldtail snakes, 54 species).

There are a handful of well-studied species within each herpetofaunal group (Fig. 3). These include the cane toad (*Rhinella marina*; 243 papers, or 29% of all papers focus on or include this species), the American bullfrog (*Lithobates catesbeianus*; 130 papers), the red-eared slider (*Trachemys scripta*; 95 papers), the brown tree-snake (*Boiga irregularis*; 57 papers), the African clawed frog (*Xenopus laevis*; 51 papers) and the brown anole (*Anolis sagrei*; 41 papers, Figs. 3 and 4). The best-studied salamander is the tiger

salamander (*Ambystoma tigrinum*, 15 papers), and the best-studied crocodilian is the spectacled caiman (*Caiman crocodilus*, 14 papers, Figs. 3 and 4). The literature on crocodiles, frogs and turtles is particularly skewed towards individual species. Over half (59%) of papers that include crocodiles focus on or include *Caiman crocodilus*, and half (50%) of the literature on frogs is focussed on or includes the cane toad *R. marina* (while *L. catesbeianus* appears in 27% of frog papers). The red-eared slider features in nearly three-quarters (73%) of literature on non-native turtles (Fig. 4). The literature on lizards is least dominated by a single taxon, with the most well-represented species, *Anolis sagrei*, appearing in 21% of papers that cover non-native lizards (Fig. 4).

## Subject focus of research

Most research (416 papers; 50%) has been conducted on impacts (Fig. 4). A large portion of this impact literature (42%), however, covers impacts of only two species, *R. marina* and *T. scripta* (Fig. 4). Similarly, the literature on control of non-native herpetofauna is heavily biased in favour of cane toads (included in 33% of papers on control) and brown tree-snakes (24% of papers on control, Fig. 4), while cane toads appear in over a third of studies (39%) on invasion correlates. There was no significant difference in the distribution of literature across subjects between papers that focussed on one species and papers that covered multiple species or which made no specific mention of species ($V = 21$, $p = 0.7422$).

## Geographic distribution of work

Excluding the 22 global studies, which largely made use of the same information base as the well-studied areas, the bulk of research on non-native herpetofauna covered in our review has been conducted in Australia (217 studies), the US (195 studies, mostly focussed on or including Florida, 86 studies and California, 41 studies), Brazil (40 studies) and Spain (40 studies), as well as several islands or island groups such as Guam (47 studies), Hawaii (46 studies) and the greater Caribbean (47 studies), with very limited information from other localities (Fig. 5). For example, the seven studies that were identified specifically from continental Africa, were all conducted in South Africa.

We further compared the geographical distribution and frequency of literature on herpetofauna to the distribution of documented introductions from *Kraus (2009)*. One-third (33%) of the ~600 species (nomenclature updated to match our list) included in the *Kraus (2009)* database have been studied in fewer localities than they have been introduced (Table 3). Excluding non-location-specific studies, only eight countries, five oceanic islands/island states and six US states have two or more location-specific studies per successful invasive amphibian or reptile introduction. The majority of localities (79%) where established populations have been recorded ($n = 191$: 145 countries and 46 US states) have fewer than two location-specific studies (aside from global or continental reviews) for every successful species (mean = 1.1), providing a poor basis for their risk analyses. Only six of the 23 localities that have more than 10 established non-native species, have more than two location-specific studies per successful

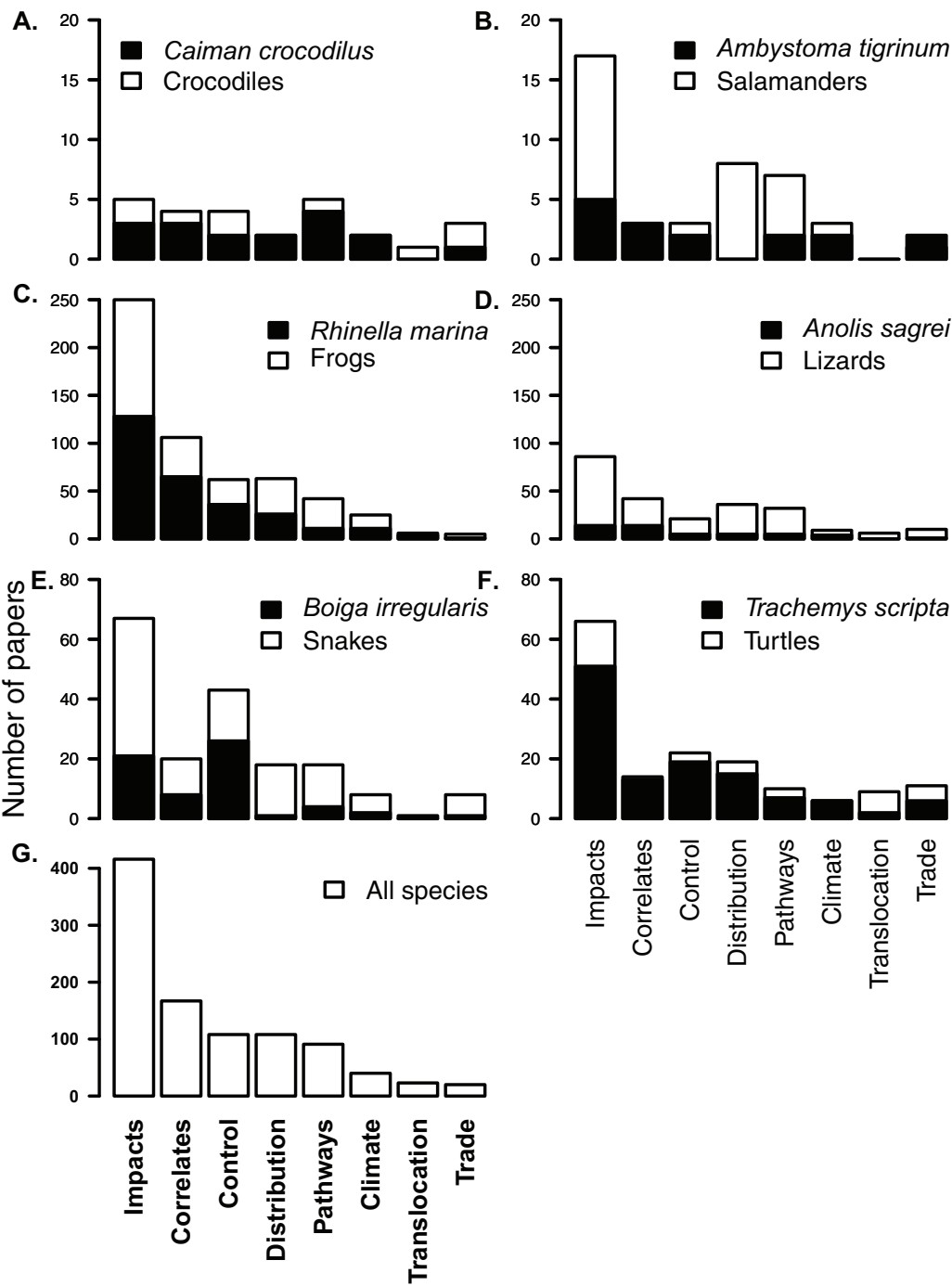

**Figure 4 Composition of subject literature on non-native amphibians and reptiles for each herpetofaunal group (A–F) and across all species (G).** In each group, papers on the most frequently studied species (A. Crocodiles *Caiman crocodilus*; B. Salamanders *Ambystoma tigrinum*; C. Frogs *Rhinella marina*; D. Lizards *Norops sagrei*; E. Snakes *Boiga irregularis*; and F. Turtles *Trachemys scripta*) are shown in black, showing that the knowledge for most non-native groups comes from a single taxon. For example, almost all papers on non-native turtles include or focus on *Trachemys scripta*. Each of the listed papers may have been included in more than one subject category. *Y*-axes for each row are different.

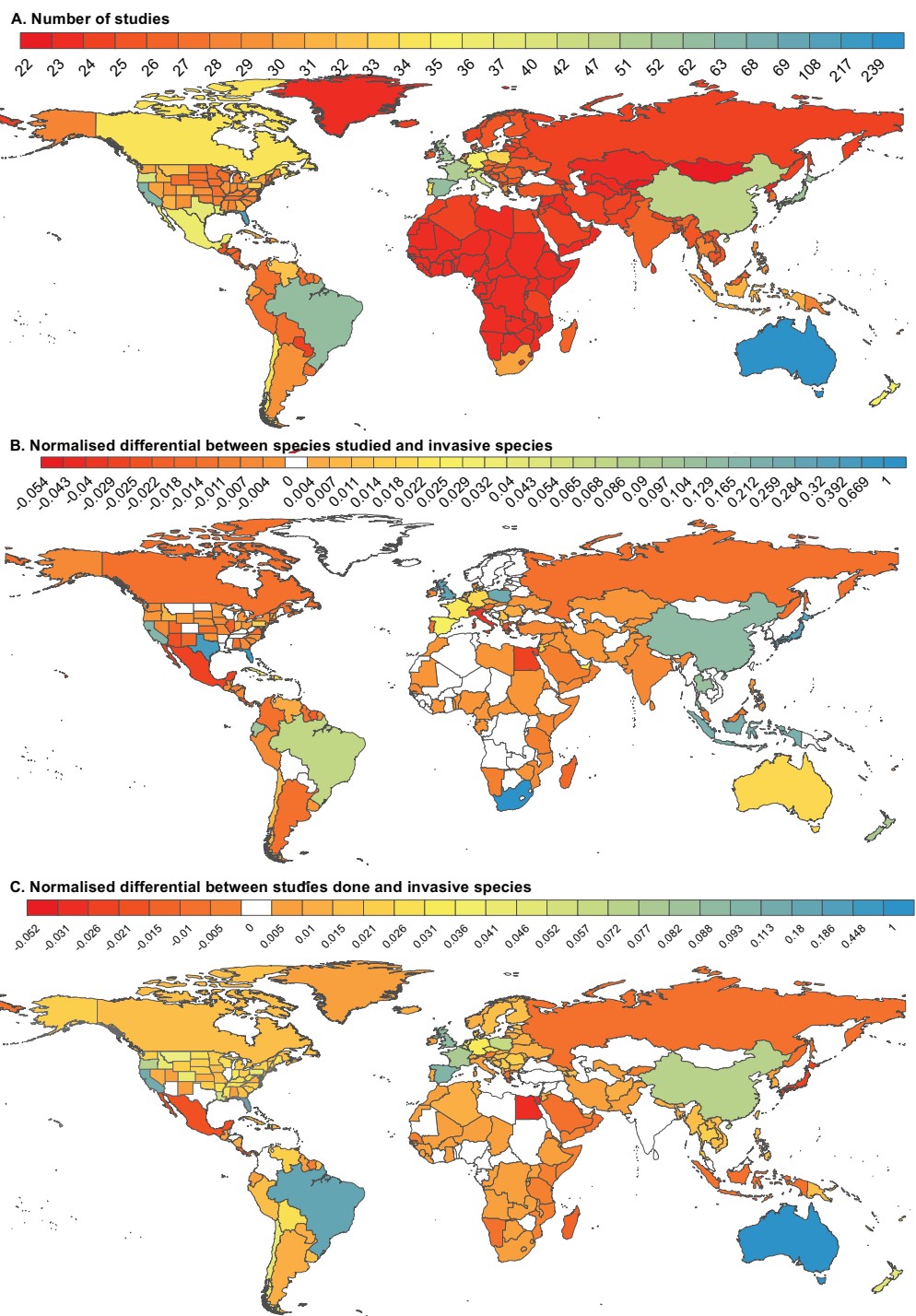

**Figure 5 The geographic distribution of studies on non-native reptile and amphibian species.**
(A) The geographic distribution of 789 studies on non-native reptile and amphibian species (20 global studies have been scored for each country and state). (B) The difference between the number of species that have been included in studies pertaining to a particular country and the number of species known to be established in that country (*Kraus, 2009*), normalised to the largest difference. (C) The difference between the number of studies conducted in a particular country and the number of species known to be established in that country (*Kraus, 2009*), normalised to the largest difference.

**Table 3 Overlap between species included in papers in the review and species documented by *Kraus (2009)*, the most comprehensive database of introduced reptiles and amphibians currently available.**

|  | Number of species in Kraus database | Percentage of taxa in Kraus | Percentage of species in Kraus |
|---|---|---|---|
| Species not identified to species level in Kraus | 65 | 10% | NA |
| Species not recorded in any of the papers in this review | 43 | 6% | 7% |
| Species in this review that only appear in global or multiregional studies | 177 | 27% | 29% |
| Species studied in more localities (states and countries) than recorded in Kraus | 107 | 16% | 18% |
| Species studied in the same number of localities as recorded in Kraus | 121 | 18% | 20% |
| Species studied in fewer localities than recorded in Kraus | 197 | 30% | 33% |
| Total species in *Kraus (2009)* | 602[1], plus 65 taxa not identified to species level | 667 | 602 |

**Note:**
[1] Another 557 taxa were listed in papers in this review but do not appear in the *Kraus (2009)* database.

species: Australia, Guam, the UK, China (inclusive of studies in Taiwan), California and France.

Additional anomalies are apparent when considering the number of species covered in the literature per location ($n = 767$ studies, Fig. 5B). In South Africa, where only seven established species have been recorded (*Kraus, 2009*), 285 species are covered by the local literature (seven studies, although the bulk of the species appear in a list of traded species), equating to 41 species documented per successfully established non-native. For localities with more than 10 established non-natives ($n = 23$), rates of study range from <0.08 species studied per successful species for La Reunion, to >3.5 species studied per successful species in Texas, the UK, Indonesia, Florida, China, Japan and the US as a whole. The US has the highest recorded number of successfully established non-natives (108 species), covered at a rate of four species studied per invasive species (451 species appear in the 192 papers that have a US focus). In the case of China (25 studies, including Taiwan, that document 47 species, at 4.3 species studied per established species), the number of successfully established species (11) is likely an underestimate (*Liu, McGarrity & Li, 2012*).

At a location level, there is clearly also selection towards studying certain species in particular places. For example, in Australia, where 28 established non-natives have been recorded (*Kraus, 2009*), 91% of studies focus on or include the cane toad. Similarly, 92% of the papers that include Guam and the Mariana Islands dealt with or include the brown tree snake, and 44% of papers that include China focus on or include the American bullfrog. Of the five most widely introduced species (*Kraus, 2009*: *T. scripta*, 84 localities (countries/US states), the Brahminy blind snake *Indotyphlops braminus*, 65 localities, *L. catesbeianus*, 58 localities, *R. marina*, 48 localities and the common house gecko *Hemidactylus frenatus*, 45 localities), only the American bullfrog—studied in 27 countries and 18 US states—has been studied in at least three quarters of the number of localities to which it has been introduced. The common house gecko (33%) and Brahminy blind snake (28%) have been studied in a third or less of the number of introduction locales.

### Success of different herpetofaunal groups

According to the data contained in *Kraus (2009)*, lizards and frogs have had the highest rates of successful establishment per species introduced (over 55% of frogs and lizards introduced outside their native range have established in at least one location), while crocodiles (at 14%) have the lowest (Table 2). However, crocodiles and turtles (~27% of known species in both groups) have had the most representative sample of species introduced from their respective herpetofaunal groups (Fig. 2). It is therefore not surprising that turtles have the most representative sample of established or invasive species: 11.5% of all described turtles have successfully established somewhere outside their native range (Table 2). The proportion of successful non-native species is much lower for other groups (typically 1–2% and as low as 0.8% for snakes, Table 2).

Despite the high rates of establishment for those lizards and frogs that have been introduced, the low representation of species that have been introduced outside of their native ranges (according to the literature on which *Kraus, 2009* is based) means that we know nothing about the invasive potential of the ~6,200 lizards and ~6,600 frogs that have never been given the opportunity to demonstrate their potential to establish or invade (Table 2).

## DISCUSSION

Our review of the herpetofaunal invasion literature identified 836 studies, covering 1,116 species. A number of distinct taxonomic, geographic and subject patterns are highlighted. Most of the literature has been produced post-2000, with a strong focus on frogs (Table 1), particularly cane toads and their impacts in Australia. While comparatively less work has been conducted on turtles and snakes, proportionately more species from both these groups have been included in studies than is the case for frogs (Fig. 2). Most countries have very little peer-reviewed literature on non-native herpetofauna (fewer than two papers per established species). Africa and Asia, in particular have had very few studies, though for southern Africa at least, this is probably a realistic reflection of the small number of introductions and invasions (*Measey et al., 2017*). Interestingly, the role of trade in the introduction of non-native species has received little attention in the period under review, despite the obvious link between trade and invasion pathways for these groups (but see *Garner et al., 2009*). Although rates of establishment success are likely inflated by higher reporting of successful introductions, the remarkably high rates of establishment success (33% for reptiles and amphibians introduced to Florida (*Krysko et al., 2016*), 25% reported for vertebrates globally (*Wilson, 2016*), and >50% for lizards and frogs (Table 2; *Kraus, 2009*)), make trade regulation and pre-border risk assessment very important management components for these species. As of 2017, research on trade of non-native herpetofauna appears to be expanding (*García-Díaz et al., 2017*; *Measey, 2017*).

Geographic and taxonomic biases in the literature are well known for most groups of invasive species (*Dawson et al., 2017*), and we expect research effort to be concentrated in areas and on species that have the biggest impacts. However, while research effort has been largely appropriate, understanding the extent of existing biases is crucial for predicting and preventing future invasions and their likely impacts, especially if for most

places and most species, the number of studies is insufficient. For example, half of the work on frogs has been conducted on the cane toad and nearly all the work on the cane toad (86%) has been conducted in Australia, meaning that despite a massive literature, this species' potential impacts on mammals, aside from marsupials, remains largely unstudied. At the same time, less well-studied species like the Asian toad (*Duttaphrynus melanostictus*) are scored as having high potential impact as a result of area-specific cultural traits, such as eating toads that has resulted in poisoning of people (see *Measey et al., 2016*), although not necessarily applicable in areas where frogs are not routinely eaten.

## Taxonomic biases

While few non-native herpetofaunal taxa have been studied in comparison to species from other groups (*Dawson et al., 2017*), the bias in studies on non-native herpetofauna is similar to the bias in information on native species. Less than half (40%) of reptile species have had their conservation status assessed (*Bland & Böhm, 2016*; *Meiri & Chapple, 2016*). Those families with the fewest conservation assessments included either families with no species identified in this review, or families that were under-represented (e.g. Amphisbaenidae). With the exception of Opluridae (Madagascan iguanas), from which no species were identified in our review, those families with the most conservation assessments (*Meiri & Chapple, 2016*) were identified to be either over-represented (Iguanidae) or proportionately represented (neither over nor underrepresented, e.g. Crotaphytidae, Phrynosomatidae) in the invasion literature. Amphibians, on the other hand, have all been assessed through The Global Amphibian Assessment (*Stuart et al., 2004*). All families identified through the global assessment to be threatened by over-exploitation (for at least one species in the family, *Stuart et al., 2004*) were also found to be well- or over-represented in the invasion literature (e.g. Ranidae, Dendrobatidae, Microhylidae, Ambystomatidae, Salamandridae, Cryptobranchidae, Fig. 2), highlighting that families or species that are used by people are more likely to become invasive.

For plants, grasses are an example of a huge family with many known invasive species. Yet only a tiny portion have been assessed to determine the extent of introduction to new localities and the level of establishment and invasion (*Visser et al., 2016*). Focussing on functional (*Canavan et al., in press*) or taxonomic groups (*Canavan et al., 2017*) has allowed scientists to distinguish syndromes of traits that enhance invasiveness. For herpetofauna, we suggest that a useful approach is to classify relevant functional groups of invaders and identify traits within these groups that correlate with invasive success (*Allen et al., 2017*; *Tingley et al., 2010*). For example, there is likely a difference in invasion correlates for species that are intentionally introduced (e.g. via the pet trade) and those that tend to move around as 'hitch-hiker' species (*Kraus, 2007*). Some groups, despite their large size, are absent from the invasion literature because no species have been introduced to localities outside their native range. For example, families that do not make good pets (e.g. snakes from the family Uropeltidae, Fig. 2) and/or are unlikely to be transported

accidentally (e.g. groups that have small, remote distributions and very specialised habitat). Other families appear to be over-represented in the invasion literature. For example, nearly all testudine families are overrepresented. This makes sense because tortoises and turtles are popular pets and are therefore widely traded and documented outside of their native ranges. The dominance of the red-eared slider in the literature, however, suggests that although testudines in general are exceptionally widely traded, proportionately fewer species actually become invasive or have notable impacts. Red-eared sliders are traded in vast numbers (common estimates are upwards of 3 million hatchlings traded globally on an annual basis, *Ramsay et al., 2007*). Reducing the volume of this trade could be achieved by enforcing stricter controls on animal husbandry and trade, and encouraging the public to be more responsible pet keepers (*Williams, 1999*).

## Geographic biases

For the large majority of countries, especially in Africa, where little information is available, there are few known established non-native species and few studies. There are, however, also countries where despite several (e.g. Egypt, Greece) or many (e.g. Japan, Indonesia) established species, very few studies have been conducted (Fig. 5C). For Japan and Indonesia, at least, the studies that have been conducted have covered a broader range of species than are known to be established (Fig. 5B). Peer-reviewed studies from these two countries, along with others like the UK, New Zealand and South Africa have covered many more species than are currently established (Fig 5B). The reason for this is not always clear, but could be due to a keen interest in herpetology in these areas or proactive research into trade and/or risk assessment (*Goka, Okabe & Takano, 2013*; *Chapple et al., 2016*).

In other locations, such as Australia and Guam, the peer-reviewed literature generation has been prolific (Fig. 5A), but focussed on single species. This means that although Australia has the most peer-reviewed literature (Fig. 5A) and the most papers produced per invasive species (Fig. 5C), most of the species that are established in Australia are still understudied (Fig. 5B). With the exception of a few places such as Florida where there is significant interest in the high numbers of invasions that appear to be driven by a large number of ports, a well-developed trade in exotic pets, and high levels of environmental disturbance (*Krysko et al., 2016*); the UK, which has a large number of English-language academic hubs; and Brazil, rates of study are worse for the rest of the globe. This likely reflects a lack of research capacity where there are invasions that are not reported or studied. For example, there are almost no studies or invasions recorded for Africa. This could be the result of fewer introductions and/or a capacity gap in herpetologists to report invasions. In other instances (e.g. China and Russia), it is possible ;that both invasions and literature on them have gone undetected because documentation regarding these events are not available in English, and therefore not searchable using the Web of Science (*Adam, 2002*). Records of first introductions for reptiles show exponential growth since about the 1950s, with no end in sight to this trend (*Seebens et al., 2017*). While there are likely already many undetected invasions,
new areas remain vulnerable and increased awareness is important for preventing future invasions.

## Setting priorities

Countries like New Zealand that have focussed on allowing importation and trade in only a short list of permitted species appear to have achieved the highest success in reducing introductions (*Genovesi et al., 2015*; *Seebens et al., 2017*). The current trend for assessing which species should be permitted on such lists is to quantify the impacts of the species based on published information. This begs the question: How many publications are needed to provide adequate information to allow for the accurate assessment of risk of invasiveness and impact? This is not an easily quantifiable number, but we can gain some insights using recent EICAT assessments and their confidence levels to suggest numbers. Without directed research, only 265 of 365 papers on 39 invasive amphibians had impacts that could be scored using the EICAT scheme (*Kumschick et al., 2017*). Of these, only eight species could be rated with high confidence for at least one impact mechanism and no species was rated with high confidence on more than one impact mechanism (EICAT scores 12 impacts, of which eight are applicable to reptiles and amphibians: Predation, Poisoning/Toxicity, Competition, Hybridisation, Disease Transmission, Interaction with other species, Parasitism, and Grazing/Herbivory/Browsing). Given that these eight species include the five most commonly studied frogs and salamanders (see Figs. 2B and 2D), we might conclude that even hundreds of studies are not sufficient to produce high-confidence scores on each impact mechanism for EICAT (ignoring SEICAT impacts). However, one study is capable of producing high confidence on one impact mechanism. Two such examples of species that were rated highly on confidence with just a single study: *Dubey, Leuenberger & Perrin (2014)* on hybridisation in Italian water frogs (*Pelophylax bergeri*) and *Holsbeek et al. (2010)* on hybridisation with Levant water frogs (*P. bedriagae*). This suggests that with more work focussed on such impact assessments, total impact for EICAT may be assessed with a minimum of eight papers, relating to each of the impact mechanisms in *Hawkins et al. (2015*, see above). To date, no EICAT assessment has found a comprehensive literature for any invasive species (*Evans, Kumschick & Blackburn, 2016*). From our work, we know that a minority of invasive herpetofaunal species have been the focus of any research. Clearly, much more directed work is needed.

Our study revealed that the majority of publications (78%) focus on single species, and that a large proportion of these, and other studies in this review (50%) concern impacts, which is good news for those hoping to score EICAT and SEICAT for these species. However, we caution that risk assessments require a fuller understanding of the invasive species, and that studies on pathways (11%) and trade (2%) are particularly poorly represented (Fig. 3). Given the importance of the first in risk assessment and the volume and key role of the latter in introductions (*García-Díaz et al., 2017*), we emphasise the need for more strategic publications analysing the trade in herpetofauna and other pathways related to their unintentional movement (*Tingley et al., 2018*).

The Global Amphibian Assessment is a good example of a world-wide initiative that drove substantial work to collate information for all species to provide a baseline and a fantastic resource for refining data. Given that only a small portion of herpetofauna currently appear to be moved around in high numbers, or show invasive tendencies, setting up more global initiatives to focus on groups of invaders across a broader geographic range is not an unrealistic task, particularly considering that distribution information is already available for amphibians. Initiatives for invaded countries to work together on the impacts of common invasive taxa could provide an important platform for accumulating crucial information on impacts. One such initiative in Europe saw members from four EU countries funded to work jointly on the impacts of the invasive African clawed frog, *X. laevis*: INVAXEN (http://www.anthonyherrel.fr/INVAXEN/). This initiative has added 15 published articles on this species, nearly doubling the available data on their invasive populations (*Courant et al., 2017*; *Louppe, Courant & Herrel, 2017*; *Rödder et al., 2017*). Funded by BIODIVERSA, this call did not include funding for studies on populations in non-participating EU countries (e.g. Italy), or in the native range of the species in southern Africa. There is scope for similar work on species like the red-eared slider, and other turtles that are currently studied in fewer locations than which they have been introduced (e.g. European pond turtles, *Emys orbicularis* and common snapping turtles *Chelydra serpentina*), several widespread gecko species, agamids like oriental garden lizards (*Calotes versicolor*), the Asian toad, and selected species from the families that are overrepresented in the literature and trade or even the cane toad outside of Australia.

Local and regional herpetological societies have a crucial role to play in this regard and should be encouraged to publish all new records of reptiles and amphibians in online databases and society websites. Many societies already do record such information in newsletters, but digitizing these data and making them available online could go a long way to improving the geographic coverage of literature and even reducing the taxonomic bias in published information. *McGeoch et al. (2016)* provide a protocol for prioritizing species, pathways and sites to assist countries in meeting Aichi Biodiversity Targets (Convention on Biological Diversity). Herpetological societies should contribute relevant information to the Global Register of Introduced and Invasive Species currently under development within the Global Invasive Alien Species Partnership framework (*McGeoch et al., 2016*). In the absence of information, risk assessments will continue to rely on information from models based on well-studied species. Improving the geographic coverage of studies on model organisms and then the taxonomic coverage of model taxa will go a long way to improving predictions for invasive species and ultimately reducing their impacts.

## ACKNOWLEDGEMENTS

R code for calculating the hypergeometric distributions was adapted from code originally written by John Wilson. We thank several reviewers who improved the clarity and content of earlier versions of this manuscript.

### Funding

This work was supported by the DST-NRF Centre of Excellence for Invasion Biology and the National Research Foundation of South Africa (grant 85417 to DMR and grant 87759 to JM). The funders had no role in study design, data collection and analysis, decision to publish, or preparation of the manuscript.

### Grant Disclosures

The following grant information was disclosed by the authors:
DST-NRF Centre of Excellence for Invasion Biology and the National Research Foundation of South Africa: 85417 to DMR and 87759 to JM.

### Competing Interests

John Measey is an Academic Editor for PeerJ.

### Author Contributions

- Nicola J. van Wilgen conceived and designed the experiments, performed the experiments, analysed the data, contributed reagents/materials/analysis tools, prepared figures and/or tables, authored or reviewed drafts of the paper, approved the final draft.
- Micaela S. Gillespie performed the experiments, approved the final draft.
- David M. Richardson conceived and designed the experiments, contributed reagents/materials/analysis tools, authored or reviewed drafts of the paper, approved the final draft.
- John Measey conceived and designed the experiments, performed the experiments, analysed the data, contributed reagents/materials/analysis tools, authored or reviewed drafts of the paper, approved the final draft.

### Data Availability

   The raw data are provided in the Supplemental Files.

### Supplemental Information

Supplemental information for this article can be found online at http://dx.doi.org/10.7717/peerj.5850#supplemental-information.

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
