# Peer review of "A taxonomically and geographically constrained information base limits non-native reptile and amphibian risk assessment: a systematic review"

_PeerJ, doi:10.7717/peerj.5850_

## Round 0.1 · original submission · Major Revisions

The reviewers and I liked the manuscript and it is an important contributions however a number of concerns have been identified around potential biases in terms of timing of the study and the words used which needed to be address before publication.

Reviewer 1 ·

Basic reporting

Basic reporting is adequate, with clear language used, sufficient field background provided, professional article structure, shared raw data and self-contained study.

Experimental design

The study represents original primary journal in line with PeerJ aims and scope. Research question is well defined, relevant and meaningful.

There are however several problems regarding the methods and study setup, and information on methods, which I presented in detail in general comments to the authors below. These issues will have to be addressed through a major revision.

Validity of the findings

Problems with the methods and interpretation of the results will have to be resolved before the findings can be judged as valid.

Conclusions are generally well stated and linked to research questions.

Additional comments

In their paper titled "A taxonomically and geographically constrained information base limits non-native species risk assessment: The case of reptiles and amphibians", MS #27799, authors presented an analysis of taxonomic attention and the presence of biases in invasion science literature.

The study is interesting ad relevant, and the manuscript is generally well developed. However, there are several problems related to the methods applied and interpretation of results, which would have to be adequately addressed before the manuscript can be considered for publishing. Please see my detailed points below.

1. My major problem with the study setup is the chosen search phrase, used for identify relevant literature. Many terms are missing that might have identified additional key literature, that is not necessarily picked up by the phrases used here - for example, terms such as newt, iguana, chameleon, agama, tuatara, gavial, gharial, skink, viper, python, serpent, alligator, caiman, etc. (even this list is far from being complete). If a publication is focused on a specific species, it is very likely that vernacular and Latin name of that species will be mentioned in the title, abstract and/or keywords (i.e. in the publication "Topic"), while general name of its taxonomic group will be often not mentioned at all, so search approach used by authors would not retrieve such publications.
I made a search in Web of Science, using the same general terms authors used (alien, invasive, non-native, exotic, non-indigenous, feral) and the specific terms I listed above: "Topic = alien OR invasive OR non-native OR exotic OR non-indigenous OR feral AND Topic = iguana* OR chameleon* OR agama* OR tuatara* OR gavial* OR gharial* OR skink* OR viper* OR python* OR serpent* OR alligator* OR caiman* OR newt*", after which I eliminated those publications identified by the search phrase made by the authors, which resulted in as much as 874 potentially relevant publications that authors failed to include in their review. Admittedly, many of these publications represent mismatches (i.e. related to medicine, like those mismatches authors identified in their sample), but the proportion of relevant publications should still be substantial.
I made also search for individual terms (together with general terms, and with exclusion of publications that were included in the sample assessed by authors) and this is what I got:
iguana* = 30
chameleon* = 26
agama* = 0
tuatara* = 5
gavial* = 0
gharial* = 0
skink* = 9
viper* = 13
python* = 53
serpent* = 262
alligator* = 151
caiman* = 4
newt* = 330
Another major problem here is that these specific search terms are likely to lead to most of the publications targeting these specific species groups, which were largely missed by the broad search phrase used by authors. As a result, it is likely that the taxonomic attention bias observed by authors is an overestimation of the true level of bias, since main body of work directed to whole such species groups is potentially missing from their dataset. This issue will have to be addressed somehow, as it can have a strong impact on results and reduce the value and validity of this study.
Best way to conduct such studies, beside searching for general species group names, is to search for specific species names (preferably Latin names), which will yield directly the value of taxonomic coverage per each species. Studied species group is large, but there are ways to conduct automatic searches, or to search for multiple species names per each search. For example, where authors provide number of papers per established species in different regions (e.g. Lines 357-358 and elsewhere in the text), it would be more appropriate to indicate the actual mean number of papers each established species really received - as it is estimated now is to an extent ambiguous and unclear.

2. Another important issue is the way papers were judged as dealing with certain species. What does it exactly mean that a species was "studied" in certain publication? This is not explained in the text, but it is critical to judge validity of the study. Was a publication considered as related to some species only if the species was really studied there (e.g fieldwork, experiment, etc.), or it was also accepted as a species-related paper if it only mentioned the species in the Introduction or Discussion (which does not seem proper to be used as a criterion)? Authors also mention in some instances that some papers that covered a lot of species represented species lists for certain country (i.e. invasive species checklist) - is that really a publication that is dealing with all those species, one that provides useful information for species assessments and the application of EICAT/SEICAT and other frameworks? This seems to be one of main weaknesses of the study, that it does not differentiate among different types of studies. A publication focused on a single species and its invasiveness will be certainly more relevant and useful for further evaluations of that species than general papers with species checklists. If nothing else, this problem should be at least discussed about in the paper, and the exclusion/inclusion criteria clearly presented.

3. The emphasis on the low proportion of the total number of species of herpetofauna that are addressed in the literature does not seem proper. It is a result that is worth to be mentioned in the manuscript, but not pushed as one of the major implications of the study (e.g. in Lines 21-22 in the Abstract and elsewhere in the text). Expectation that all of the species (even all those that were never recorded and not expected to be recorded out of their home range) is unrealistic, especially bearing in mind limited research funds that are not easy to obtain even for actual invasive species, with proven negative impacts. While general studies that assess invasion potential across all species groups, i.e. based on the general life history traits, should be certainly encouraged, it is not likely that a single-species study focusing on hypothetical invasion risk of some exclusively native species would be approved for funding, or be judged positively by editors of scientific journals. I would suggest keeping this issue less emphasized in the text, and instead to focus on and assess proportion of covered species to the total number of species that were confirmed as alien or invasive. That will be a much more meaningful indication of a true level of bias.

4. When discussing about poor coverage of species in the literature from most of the countries, especially those from non-English speaking countries, authors should address the issue of biased coverage of the database used in the analysis, with a strong English literature bias. Web of Science has a strong bias against non-English literature, which limits the usefulness of such databases for this type of assessments (e.g. Adam et al. 2002; Zetterström 2002). Majority of literature in many countries gets published in journals that are not indexed by WoS, while still being locally highly relevant, available and readily used for invasive species risk screening and impact assessments. As such, species coverage indicated by Web of Science and apparent biases will be of somewhat limited value, and should be at least properly noted in the study. This is briefly mentioned in Lines 448-449, but it deserves more attention, including the Methods section (i.e. a paragraph describing the appropriateness and limitations of the used source of data), as it represents one of the major limitations of the study.

5. Regarding different countries and publishing intensity regarding hepetofauna, it would be useful if the total number of publications on herpetofauna per each country would be compared with the total number of publications from the field of invasion science in general per country (e.g. obtained with the general search phrase "Topic = alien OR invasive OR non-native OR exotic OR non-indigenous OR feral"). That would provide a good indication of differences among countries, i.e. which countries proportionally publish more papers on invasive herpetofauna.

6. Lines 25-26, "Less work is published on turtles and snakes, but proportionately more species from both these groups have been studied than for frogs." - this is a confusing sentence, consider revising it. Statement also appears contradictory to the previous sentence (i.e. "strong focus on frogs (58%)").

7. Line 69, "more expensive" - consider rephrasing to avoid confusion (it could be understood as more expensive for purchase), for example "more costly management" or something in that sense. Also, in the same Line, replace "venemous" with "venomous"

8. Line 110 - use passive tense; replace "perform" with "performed"

9. Line 250 - replace "species" with "publications" or "studies"

10. Lines 280-282 and elsewhere in the text - authors refer to particular species interchangeably by their Latin and vernacular name. They should provide both where the species is mentioned for the first time in the text, and afterwards consistently use only one of them, either Latin or vernacular.

11. Lines 308-320 - this section is confusing and hard to follow (e.g. "seven studies covering 285 species for only seven successfully established non-natives", "species studied per successful species", etc.). It should be revised so it is easier to understand this part of results.

12. In line 311-312, authors state "although the bulk of these are lists of traded species" - as discussed above, it is questionable whether these are really papers dealing with invasive species, if they only study species trade, as well as whether these species were indeed "studied", i.e. whether such paper should be counted for those species.

13. Line 322 "There are also strong biases in the number of papers per invasive species." - Authors discuss here about hte high number of studies per cane toad, and brown tree snakes. Are such publication patterns really biases? If these species have such strong negative impacts, high attention and number of publications is not really inappropriate. If brown tree snake is such a sever pest in Guam, it is only natural that the major focus will be on that species, and it is not to be expected (or appropriate) that the authors from Guam will instead use funds and their time to study instead another species that was never introduced there or elsewhere. My point is that authors should be consider what are real biases and what are appropriate allocations of research effort. This also pertains to other sections of the paper.

14. Authors sometimes refer to "regions" (Line 312) and sometimes to "localities" (Line 327), which is a bit confusing.

15. Line 364 - replace "herps" with a proper term, e.g. "reptiles and amphibians"

16. Line 369 - I am not sure about the PeerJ guidelines, but "Online early" articles (i.e. those not yet assigned to an issue) should be cited as "in press", not "Online early".

17. Lines 460-462, "This begs the question: How many publications are needed to provide adequate information to allow for the accurate assessment of risk of invasiveness and impact?" - the question is incomplete, as it is not just matter of the number of publications, but quality too. Some of the paper types included as studies of certain species, such as simple alien or traded species lists (see my comments above) can not be considered as equally beneficial for this process as focused single-species studies.

18. References in Lines 474, 475, 478 and 517 are cited with "et al." placed in Italics, while it is not so in other sections of the text. Furthermore, Dubey et al. 2014 publication has three-authors, and such publications are elsewhere cited with all authors' names listed.

19. All references cited in the text are included in the reference list, and vice versa. Reference list should be however checked for formatting, e.g. capitalization of journal name in Lines 694-695, inconsistent referring to the R programming language in the text (Line 226) and reference list (Line 706).


References:

Adam, D. (2002). The counting house. Nature 415, 726-729.

Zetterström, R. (2002). Bibliometric data: a disaster for many non-American biomedical journals. Acta Paediatrica 91, 1020-1024.

Reviewer 2 ·

Basic reporting

I think that the manuscript could improve if the authors provide a bit more of context. However, I don't think that it fails to meet PeerJ's standards, and it is not something that, by itself, should prevent the publication of this manuscript. I have provided more detailed comments in my 'general comments to the authors'

Experimental design

No comment

Validity of the findings

The results reported in the manuscript and, by extension, the conclusions are based on a systematic review conducted two years ago (line 128; date: 03/03/2016). The field of amphibian and reptile invasion ecology has seen some progress since then, and I am concerned that the authors are missing some key references important to assess the current state of the knowledge on the topic. Please, see my detailed feedback in the 'general comments to the authors' section

Additional comments

Thanks for giving me the opportunity to read and review the manuscript ‘A taxonomically and geographically constrained information base limits non-native species risk assessments: The case of reptiles and amphibians’ by van Wilgen et al. A systematic review of the literature on alien amphibians and reptiles is timely to synthesise the progress of the field, which has advanced largely thanks to Kraus’ 2009 book. It is also a key step towards identifying potential knowledge gaps to guide future research and management efforts to address the increasing risks posed by alien amphibians and reptiles. The methods used by the authors are appropriate and follow the guidelines for conducting literature reviews. The presentation of the results is, in general, good, although I think the manuscript will benefit from a better statement of the objectives and purposes of the review, as well as the implications for the management of alien amphibians and reptiles.
Unfortunately, the results reported in the manuscript and, by extension, the conclusions are based on a systematic review conducted two years ago (line 128; date: 03/03/2016). The field of amphibian and reptile invasion ecology has seen some progress since then, and I am concerned that the authors are missing some key references important to assess the current state of the knowledge on the topic. For example, I was rather surprised to find no reference to Capinha et al., (2017)(see references cited below), who revised and updated Kraus’ 2009 database on the global distribution of established alien amphibians and reptiles. The authors themselves recognise that Kraus book, although a landmark work in the area of alien amphibians and reptiles, is probably dated. I am aware that Capinha et al. publication, unlike Kraus 2009 book, does not include introduced but not established species Nonetheless, I don’t think this is a major impediment for the authors to use that newer database. Capinha’s research paper presented a substantial update on the global distribution of established alien amphibians and reptiles. Given how much the authors rely on Kraus’ 2009 database for supporting their results and conclusions, I am wondering how much their results and conclusions would change if they were using Capinha’s database. There are other relevant papers on alien amphibians and reptiles published since 2016 that the author’s systematic review does not include, even if cited in the main text of the manuscript. Examples include (Tingley et al., 2016a; Allen et al., 2017; García-Díaz et al., 2017b; Li et al., 2016; Tingley et al., 2016b, 2017; Capinha et al., 2017; Tingley et al., 2018; Filz et al., 2017; García-Díaz et al., 2017a; Forti et al., 2017; Héritier et al., 2017; Vences et al., 2017; Rosa et al., 2018). I wonder how the inclusion of these additional references in their systematic review, even if they seem to show similar geographical patterns as those reported in the manuscript, would change the results and conclusions reached in the manuscript.
In summary, I am concerned that the results and conclusions presented in the manuscript may not reflect the current state of our knowledge about alien amphibians and reptiles. I do understand that the editorial procedures and decisions made by the journal to where this manuscript was submitted previously are partly to blame for this issue, but it is important that I point this out.

I have also read through the Excel spreadsheet provided by the authors (Supplemental Material), and I realised that there are a couple of interesting papers on the ecology and management of alien amphibians and reptiles published in the period considered by the authors that were not included in their review. For example: (Fujisaki et al., 2015; García-Díaz & Cassey, 2014; Kopecký et al., 2013). Moreover, I wonder whether using only English searching terms and restricting the literature search to the Web of Science (my experience is that Google Scholar is better for finding grey literature, which composes a non-negligible share of the research on alien amphibians and reptiles) may have constrained the range of publications that the authors were able to find. However, I want to emphasise that I don’t think that this is a necessarily important issue in the manuscript – the main goal of a systematic review is to obtain a representative sample and the procedures followed by the authors are sound in this respect.

More generally, I think that the manuscript will benefit from a clearer objective and purpose and, concomitantly, from a better linkage between the research presented and the conclusions and management recommendations. Currently, the manuscript focuses on making a case for showing how the scarcity of studies could hinder the application of risk assessment tools that assess impacts of alien species, such as the (S)EICAT. This is fine for the case of prioritising actions to tackle alien amphibians and reptiles that are already established, as the authors state it in lines 77-79. However, in other parts of the manuscript, the authors suggest that a lack of information may also hinder risk assessments to inform preventive strategies (for example, lines 79-80 and the section ‘Setting priorities’ in the discussion). The later is the main formal application of risk assessments for alien amphibians and reptiles in most jurisdictions. Data-intensive approaches defeat the purpose of risks assessments in informing preventive strategies, particularly considering the pace at which new species of alien amphibians and reptiles are being introduced. Should we wait until they are established somewhere, and we have enough research to manage them? Moreover, I suggest that species-based management measures, like those explained by the authors, are effective when the identities of the species are known. Thus they are appropriate for prioritising established species and pathways such as the legal wildlife trade. However, lots of alien amphibians and reptiles are transported unintentionally, and their identities are not necessarily known a priori, and pathway-level measures would be better suited as preventive strategies in those circumstances (McGeoch et al., 2015; García-Díaz et al., 2017b; Leung et al., 2014). In summary, I suggest that the conclusions presented in the manuscript will benefit from setting the context in which the recommendations would be applicable. Perhaps it could be interesting to evaluate the implications of the results in the context of the different stages of the invasion pathway and different transport pathways. The authors briefly talk about this in lines 486-489, but I think it would be good to expand the discussion on this topic.

Eliciting biases in the literature depend on making comparisons against null expectations. I have read the manuscript a couple of time, and I think the methodological procedures are sound, but I suggest that the authors describe their expectations more explicitly – that will certainly help to understand their results and to showcase current knowledge gaps. Why not, for example, partition the results explicitly by stage of the invasion pathway? I don’t necessarily expect the number of studies of impacts of alien species to correlate with the number of species introduced into a region (i.e., successful and failed introductions). However, I kind of expect that the number of studies will increase with the number of established species (the authors have done this analysis). The authors have the data necessary for the analysis by invasion pathway stage, and I think it would be interesting.

Those were my major comments to the authors. I have some other minor suggestions:

It would be good that the authors define ‘alien’ and ‘invasive’ early on in the introduction, the first time they are mentioned. Currently, these terms are defined for the first time in the methods section, and the introduction makes it sound as if they are interchangeable.

Line 67: the importance of release numbers was also reported by Mahoney et al., (2014).

Line 70: considering recent research (Allen et al., 2017; García-Díaz et al., 2016; Mahoney et al., 2014), I would say that it is not only early maturity but fast-paced life histories in general.

Lines 73-74: I disagree with the statement that the predictive ability of models is unsatisfactory. In fact, I would argue it the predictive abilities of models assessing the probability of establishment are rather good, and models are being used in a number of places to inform management. See for example (Bomford et al., 2005; Springborn et al., 2011; García-Díaz et al., 2016; Kopecký et al., 2013). It is, of course, a different matter for models attempting to predict impacts, but that is not the topic of the reference cited at the end of the sentence in the manuscript.

Lines 367-369: Tingley et al. is not a paper on the intentional trade of herpetofauna that is being discussed here. It is a work on the unintentional transport of toads hitchhiking in goods and commodities. The trade on those commodities results in the transport of the toads, but the toads are not traded.

References cited
Allen W. L., Street S. E. & Capellini I. (2017) Fast life history traits promote invasion success in amphibians and reptiles. Ecology Letters 20: 222–230
Bomford M., Kraus F., Braysher M., Walter L. & Brown L. (2005) Risk assessment model for the import and keeping of exotic reptiles and amphibians. Canberra, Australia: Bureau of Rural Sciences
Capinha C., Seebens H., Cassey P., García-Díaz P., Lenzner B., Mang T., Moser D., Pyšek P., Rödder D., Scalera R., Winter M., Dullinger S. & Essl F. (2017) Diversity, biogeography and the global flows of alien amphibians and reptiles. Diversity and Distributions 23: 1313–1322
Filz K. J., Bohr A. & Lötters S. (2017) Abandoned Foreigners: is the stage set for exotic pet reptiles to invade Central Europe? Biodiversity and Conservation: 1–19
Forti L. R., Becker C. G., Tacioli L., Pereira V. R., Santos A. C. F., Oliveira I., Haddad C. F. & Toledo L. F. (2017) Perspectives on invasive amphibians in Brazil. PloS one 12: e0184703
Fujisaki I., Mazzotti F. J., Watling J., Krysko K. L. & Escribano Y. (2015) Geographic risk assessment reveals spatial variation in invasion potential of exotic reptiles in an invasive species hotspot. Herpetological Conservation and Biology 10: 621–632
García-Díaz P. & Cassey P. (2014) Patterns of transport and introduction of exotic amphibians in Australia. Diversity and Distributions 20: 455–466
García-Díaz P., Ramsey D. S. L., Woolnough A. P., Franch M., Llorente G. A., Montori A., Buenetxea X., Larrinaga A. R., Lasceve M., Álvarez A., Traverso J. M., Valdeón A., Crespo A., Rada V., Ayllón E., Sancho V., Lacomba J. I., Bataller J. V. & Lizana M. (2017a) Challenges in confirming eradication success of invasive red-eared sliders. Biological Invasions 19: 2739–2750
García-Díaz P., Ross J. V., Woolnough A. P. & Cassey P. (2016) The illegal wildlife trade is a likely source of alien species. Conservation Letters 10: 690–698
García-Díaz P., Ross J. V., Woolnough A. P. & Cassey P. (2017b) Managing the risk of wildlife disease introduction: pathway-level biosecurity for preventing the introduction of alien ranaviruses. Journal of Applied Ecology 54: 234–241
Héritier L., Valdeón A., Sadaoui A., Gendre T., Ficheux S., Bouamer S., Kechemir-Issad N., Du Preez L., Palacios C. & Verneau O. (2017) Introduction and invasion of the red-eared slider and its parasites in freshwater ecosystems of southern Europe: risk assessment for the European pond turtle in wild environments. Biodiversity and Conservation: 1–27
Kopecký O., Kalous L. & Patoka J. (2013) Establishment risk from pet-trade freshwater turtles in the European Union. Knowledge and Management of Aquatic Ecosystems: 02
Leung B., Springborn M. R., Turner J. A. & Brockerhoff E. G. (2014) Pathway-level risk analysis: the net present value of an invasive species policy in the US. Frontiers in Ecology and the Environment 12: 273–279
Li X., Liu X., Kraus F., Tingley R. & Li Y. (2016) Risk of biological invasions is concentrated in biodiversity hotspots. Frontiers in Ecology and the Environment 14: 411–417
Mahoney P. J., Beard K. H., Durso A. M., Tallian A. G., Long A. L., Kindermann R. J., Nolan N. E., Kinka D. & Mohn H. E. (2014) Introduction effort, climate matching and species traits as predictors of global establishment success in non-native reptiles. Diversity and Distributions 21: 64–74
McGeoch M. A., Genovesi P., Bellingham P. J., Costello M. J., McGrannachan C. & Sheppard A. (2015) Prioritizing species, pathways, and sites to achieve conservation targets for biological invasion. Biological Invasions 18: 299–314
Rosa C. A. da, Zenni R., Ziller S. R., Curi N. de A. & Passamani M. (2018) Assessing the risk of invasion of species in the pet trade in Brazil. Perspectives in Ecology and Conservation 16: 38–42
Springborn M., Romagosa C. M. & Keller R. P. (2011) The value of nonindigenous species risk assessment in international trade. Ecological Economics 70: 2145–2153
Tingley R., García-Díaz P., Arantes C. R. R. & Cassey P. (2018) Integrating transport pressure data and species distribution models to estimate invasion risk for alien stowaways. Ecography 41: 635–646
Tingley R., Mahoney P. J., Durso A. M., Tallian A. G., Morán-Ordóñez A. & Beard K. H. (2016a) Threatened and invasive reptiles are not two sides of the same coin. Global Ecology and Biogeography 25: 1050–1060
Tingley R., Thompson M. B., Hartley S. & Chapple D. G. (2016b) Patterns of niche filling and expansion across the invaded ranges of an Australian lizard. Ecography 39: 270–280
Tingley R., Ward-Fear G., Schwarzkopf L., Greenlees M. J., Phillips B. L., Brown G., Clulow S., Webb J., Capon R., Sheppard A., Strive T., Tizard M. & Shine R. (2017) New Weapons in the Toad Toolkit: A Review of Methods to Control and Mitigate the Biodiversity Impacts of Invasive Cane Toads (Rhinella Marina). The Quarterly Review of Biology 92: 123–149
Vences M., Brown J. L., Lathrop A., Rosa G. M., Cameron A., Crottini A., Dolch R., Edmonds D., Freeman K. L. & Glaw F. (2017) Tracing a toad invasion: Lack of mitochondrial DNA variation, haplotype origins, and potential distribution of introduced Duttaphrynus melanostictus in Madagascar. Amphibia-Reptilia 38: 197–207

External reviews were received for this submission. These reviews were used by the Editor when they made their decision, and can be downloaded below.

---

## Round 0.2 · accepted · Accept

Thanks you for considering the comments of the reviewers. As there is a clear misunderstanding regarding the purpose and how to undertake a systematic evidence review I have gone through it myself. I am happy to say that I am satisfied with your responses and the corrections you have made to the paper and am therefore happy to recommend it be accepted.

As a note in response to a comment in the rebuttal letter I don't think I have given my name to a review I have undertaken.

# External reviews were received for this submission. These reviews were used by the Editor when they made their decision, and can be downloaded below.